# Leucine alleviates cytokine storm syndrome by regulating macrophage polarization via the mTORC1/LXRα signaling pathway

Hui Yan*†, Yao Liu†, Xipeng Li†, Bing Yu, Jun He, Xiangbing Mao, Jie Yu, Zhiqing Huang, Yuheng Luo, Junqiu Luo, Aimin Wu, Daiwen Chen*

Key Laboratory of Animal Disease Resistance Nutrition of China Ministry of Education, Key Laboratory of Animal Disease resistant Nutrition and Feed of China Ministry of Agriculture and Rural Affairs, Key Laboratory of Animal Disease resistant Nutrition of Sichuan Province, Animal Nutrition Institute, Sichuan Agricultural University, Chengdu, China

**\*For correspondence:**
yan.hui@sicau.edu.cn (HY);
dwchen@sicau.edu.cn (DC)

†These authors contributed equally to this work

**Competing interest:** The authors declare that no competing interests exist.

**Abstract** Cytokine storms are associated with severe pathological damage and death in some diseases. Excessive activation of M1 macrophages and the subsequent secretion of pro-inflammatory cytokines are a major cause of cytokine storms. Therefore, promoting the polarization of M2 macrophages to restore immune balance is a promising therapeutic strategy for treating cytokine storm syndrome (CSS). This study was aimed at investigating the potential protective effects of leucine on lipopolysaccharide (LPS)-induced CSS in mice and exploring the underlying mechanisms. CSS was induced by LPS administration in mice, which were concurrently administered leucine orally. In vitro, bone marrow derived macrophages (BMDMs) were polarized to M1 and M2 phenotypes with LPS and interleukin-4 (IL-4), respectively, and treated with leucine. Leucine decreased mortality in mice treated with lethal doses of LPS. Specifically, leucine decreased M1 polarization and promoted M2 polarization, thus diminishing pro-inflammatory cytokine levels and ameliorating CSS in mice. Further studies revealed that leucine-induced macrophage polarization through the mechanistic target of rapamycin complex 1 (mTORC1)/liver X receptor α (LXRα) pathway, which synergistically enhanced the expression of the IL-4-induced M2 marker Arg1 and subsequent M2 polarization. In summary, this study revealed that leucine ameliorates CSS in LPS mice by promoting M2 polarization through the mTORC1/LXRα/Arg1 signaling pathway. Our findings indicate that a fundamental link between metabolism and immunity contributes to the resolution of inflammation and the repair of damaged tissues.

## eLife assessment

The study has added value to what we have already known in the potential pharmacological immunomodulatory therapies in LPS-induced sepsis, and especially the use of oral leucine might be of great interest to the readers engaged in this field. We believe this study is **important** and provides **solid** evidence on the potential use of leucine in sepsis.

## Introduction

Cytokine storm syndrome (CSS), characterized by severe systemic inflammation, is the leading cause of death in certain infectious diseases, such as Coronavirus disease 2019 and acute respiratory distress

syndrome (*Kim et al., 2021*; *Ramasamy and Subbian, 2021*; *Wang et al., 2020*). Acute overwhelming inflammation caused by CSS is characterized by elevated levels of circulating cytokines and hyperactivation of immune cells, particularly macrophages (*Grom et al., 2016*). Macrophage hyperactivation markedly increases the levels of many pro-inflammatory cytokines, such as interleukin-1beta (IL-1β), interleukin-6 (IL-6), tumor necrosis factor alpha (TNF-α), and interferon gamma (IFN-γ), thereby leading to a hyperinflammatory state associated with high mortality (*Schulert and Grom, 2015*). Macrophages are broadly classified into two categories according to their function: M1 (classical) and M2 (alternative). M1 triggers a rapid pro-inflammatory response to infection and tissue damage, whereas M2 exhibits anti-inflammatory and reparative activity, participating in inflammation remission and tissue repair (*Covarrubias et al., 2015*; *Ginhoux et al., 2016*; *Murray et al., 2014*). Therefore, targeting macrophages to achieve a precise balance of anti- and pro-inflammatory activity is a major avenue for the treatment of CSS. Currently, the main approach to treating CSS involves mitigating excessive immune responses through broad (e.g., glucocorticoid) and targeted (e.g., anti-cytokine) approaches (*Schulert and Grom, 2014*). However, those approaches elicit numerous adverse effects, such as hypertension and liver damage. Nutritional intervention is a relatively safe approach to alleviate CSS.

Macrophage polarization to M1 or M2 phenotypes has distinct metabolic requirements, in which the mechanistic target of rapamycin complex 1 (mTORC1) pathway plays a key role (*Kang et al., 2018*). mTORC1 is an important metabolic regulator that controls cell proliferation, differentiation, and immunity by sensing cellular energy status, nutrients, and external stimuli (*Dibble et al., 2012*). Amino acid restriction reprograms macrophage function through an mTOR-centric cascade (*Orillion et al., 2018*). Leucine, an important essential amino acid, is also a critical mTORC1 regulator (*Jewell et al., 2015*). The mTORC1 pathway plays a key role in controlling macrophage polarization (*Byles et al., 2013*). Recent studies have indicated that leucine blocks macrophage infiltration in obese animals, thus decreasing levels of the pro-inflammatory cytokine TNF-α and the macrophage marker F4/80[+] in adipocytes (*Macotela et al., 2011*). However, the mechanism through which leucine alleviates inflammation is unclear.

In our study, we explored the effects of leucine on lipopolysaccharide (LPS)-induced CSS and elucidated the essential roles of leucine in CSS pathogenesis. We observed that leucine decreased mortality in mice treated with lethal doses of LPS. In addition, leucine decreased the expression and secretion of inflammatory factors in the serum and tissues of mice. In vitro data further confirmed that leucine inhibited LPS-driven M1 polarization and promoted M2 polarization through the mTORC1 signaling pathway, and leucine promoted M2 macrophage polarization through enhancing the expression of liver X receptor α (LXRα) induced by IL-4. Thus, our findings revealed a basic link between metabolism and immunity, in which leucine, via mTORC1/LXRα/Arg1, regulates the polarization of macrophages to M2 and alleviates inflammation.

## Results

### Leucine improves survival and inhibits cytokine storm in LPS-induced endotoxemic mice

Firstly, we determined whether leucine prevents LPS-induced acute endotoxemia by i.p. injection of 25 mg/kg LPS. The injection of a lethal dose of LPS caused the death of all mice within 72 hr. Providing 2% leucine from feed or 5% leucine from water improved survival rate of LPS-injected mice to 34% and 60%, respectively (*Figure 1A*). Thus, leucine appeared to protect against LPS-induced acute endotoxemia in mice.

LPS-induced endotoxemia and death are partially due to systemic and local inflammation. We next investigated the effect of leucine on acute inflammation caused by non-lethal doses of LPS in mice. Mice fed leucine in the feed, drinking water, or a combination of both exhibited higher body weight gain than control mice before LPS stimulation (*Figure 1B*). In addition, LPS, compared with the control, significantly increased levels of the pro-inflammatory cytokines IL-6, IFN-γ, and TNF-α in both the serum and liver after 6 hr i.p. injection. Mice administered 2% leucine in the feed, 5% leucine in the drinking water, or a combination of both, compared with the LPS group, showed significantly lower cytokine levels in the serum and liver (*Figure 1C,D*). We also investigated expression of inflammation-related markers in various tissues. Compared with that in the LPS group, the expression of *Il6*, *Il1b*, *Nlrp3*, *Mcp1*, and *Inos* in liver and spleen was significantly lower in mice receiving 2% leucine from the

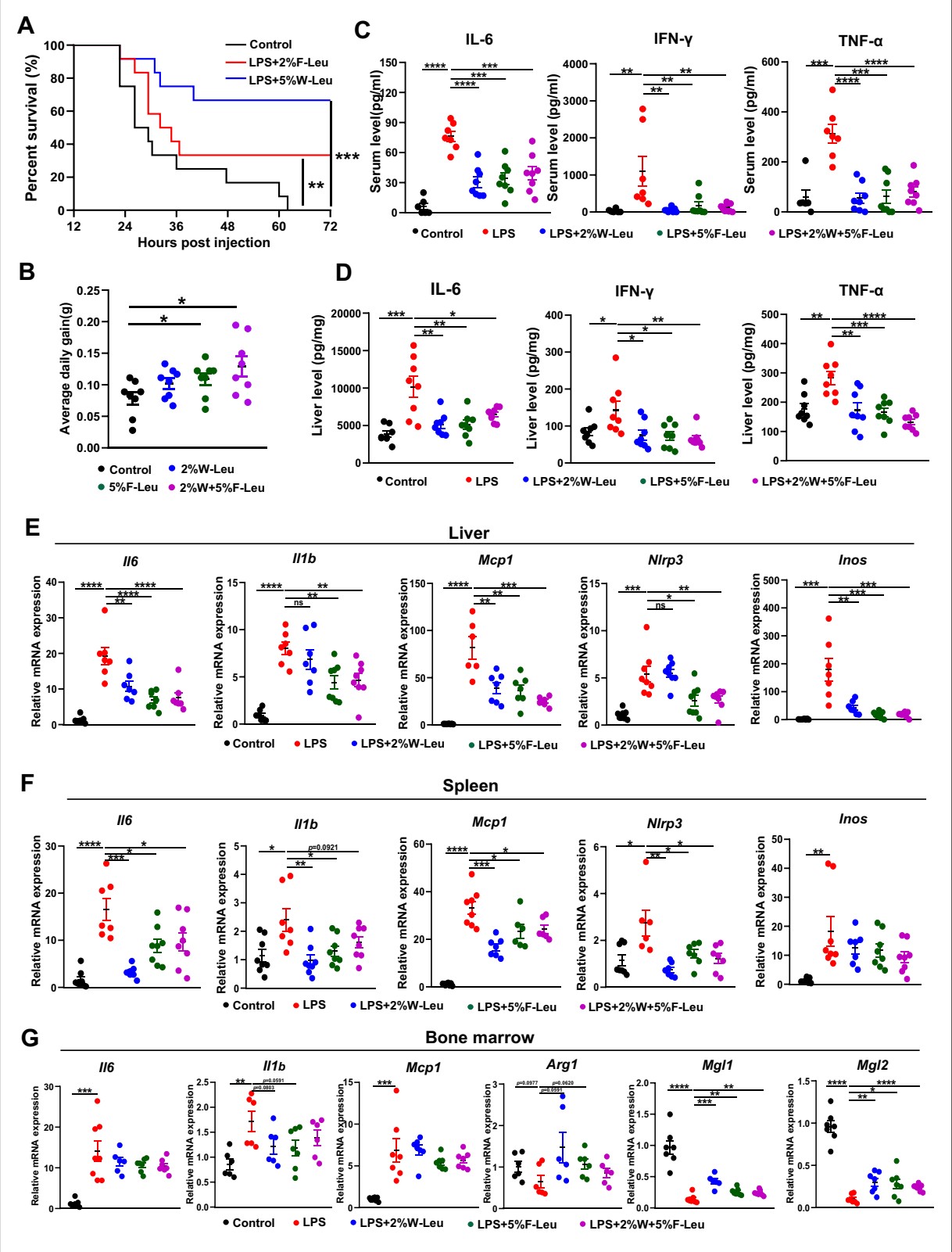

**Figure 1.** Leucine ameliorates lipopolysaccharide (LPS)-induced inflammation. (**A**) Kaplan–Meier curve showing survival of the mice (*n* = 12). (**B**) Average daily weight gain of mice (*n* = 8). (**C, D**) Measurement of IL-6, IFN-γ, and TNF-α secretion in mouse serum and liver by enzyme-linked immunosorbent assay (ELISA) after treatment with LPS for 6 hr. (**E–F**) mRNA expression of Il6, Il1β, Nlrp3, Mcp1, and Inos, measured by real-time PCR in the liver and

*Figure 1 continued on next page*

*Figure 1 continued*

spleen. (**G**) mRNA expression of *Il6*, *Il1b*, *Mcp1*, *Arg1*, *Mgl1*, and *Mgl2*, measured by real-time PCR in the bone marrow. Student's *t*-test was used to determine statistical significance, defined as *p < 0.05, **p < 0.01, ***p < 0.001, and ****p < 0.0001.

The online version of this article includes the following source data for figure 1:

**Source data 1.** Mouse survival curves (*Figure 1A*).

**Source data 2.** Mouse body weight (*Figure 1B*).

**Source data 3.** Mouse serum inflammatory levels (*Figure 1C*).

**Source data 4.** Mouse liver inflammatory levels (*Figure 1D*).

**Source data 5.** mRNA expression of genes in liver (*Figure 1E*), spleen (*Figure 1F*), and bone marrow (*Figure 1G*).

feed, 5% leucine from the drinking water, or a combination of both (*Figure 1E–G*). In addition, mice receiving 2% leucine from the feed, 5% leucine from the drinking water, or a combination of both, compared with the LPS group, showed lower expression of *Il6*, *Il1b*, and *Mcp1*, and higher expression of the anti-inflammation-related markers *Arg1*, *Mgl1*, and *Mgl2* in the bone marrow (*Figure 1E–G*).

Collectively, our results indicated that 2% leucine from feed, 5% leucine from drinking water, or a combination of both significantly inhibits pro-inflammatory cytokine production in mice with LPS-induced endotoxemia.

## Leucine regulates macrophage polarization in endotoxemic mice

Inflammatory cytokines are derived primarily from a variety of immune cells, such as macrophages and neutrophils. Briefly, LPS i.p. injection resulted in significantly more white blood cells, neutrophils, monocytes, eosinophils, and basophils in the blood than observed in the control group. Providing leucine in the drinking water, feed, or both significantly decreased white blood cells, neutrophils, monocytes, eosinophils, and basophils (*Figure 2A*). Thus, the alleviation of inflammation and death in CSS might have been due to changes in immune cell populations and the regulation of immune cell differentiation by leucine.

Macrophages are derived from monocytes and are the main cells producing cytokines. Typically, macrophages polarize to pro-inflammatory (M1) and anti-inflammatory (M2) phenotypes, depending on their microenvironment (*Locati et al., 2020*). Therefore, we determined how leucine modulates macrophage polarization by performing flow cytometry sorting in endotoxemic mice. The immune cell population was labeled by CD45$^+$, and CD11b$^+$ and F4/80$^+$ double-positive labeled macrophages were used. To determine the proportions of macrophage subsets, we used CD86$^+$ and CD206$^+$ as molecular markers for M1 and M2 macrophages, respectively (*Figure 2B*). Briefly, LPS i.p. injection significantly increased the percentage of CD86$^+$ and decreased the percentage of CD206$^+$ in both the bone marrow and spleen, thus suggesting that LPS i.p. injection led to a transition toward M1 polarization in mice (*Figure 2C,D*). In contrast, providing leucine in the drinking water, feed, or both decreased the percentage of CD86$^+$ and increased the percentage of CD206$^+$, thus indicating that leucine ameliorated inflammation in mice and decreased macrophage M1 polarization, but markedly promoted M2 polarization (*Figure 2C,D*). Together, these results indicated that the anti-inflammatory effects of leucine were probably mediated by modulating macrophage polarization, through suppressing M1 polarization but promoting M2 polarization.

To further evaluated the role of macrophage in leucine-mediated alleviation of CSS, macrophages were depleted in mice by vein injection of clodronate-containing liposomes before administration of LPS. Macrophage depletion significantly eliminated the mitigating effect of leucine in endotoxemic mice by blood immune cells (*Figure 2E*) and serum inflammatory factors IL-6, TNF-α, and IFN-γ (*Figure 2F*).

## Leucine promotes M2 polarization in BMDMs

To further determine the effects of leucine on macrophage polarization, we induced BMDMs to differentiate into M1 or M2 macrophages through stimulation with LPS or IL-4, respectively (*Figure 3A*). LPS promoted M1 macrophage polarization, as indicated by increased release of the pro-inflammatory cytokines IL-6 and TNF-α, and induction of mRNA expression of *Il1β*, *Tnfa*, *Il6*, and *Nlrp3* in BMDMs (*Figure 3B, C*). In LPS-stimulated cells, leucine decreased the secretion of IL-6 and TNF-α in culture supernatant and suppressed the expression of *Il1β* and *Tnfα*, particularly at 10 mM concentration

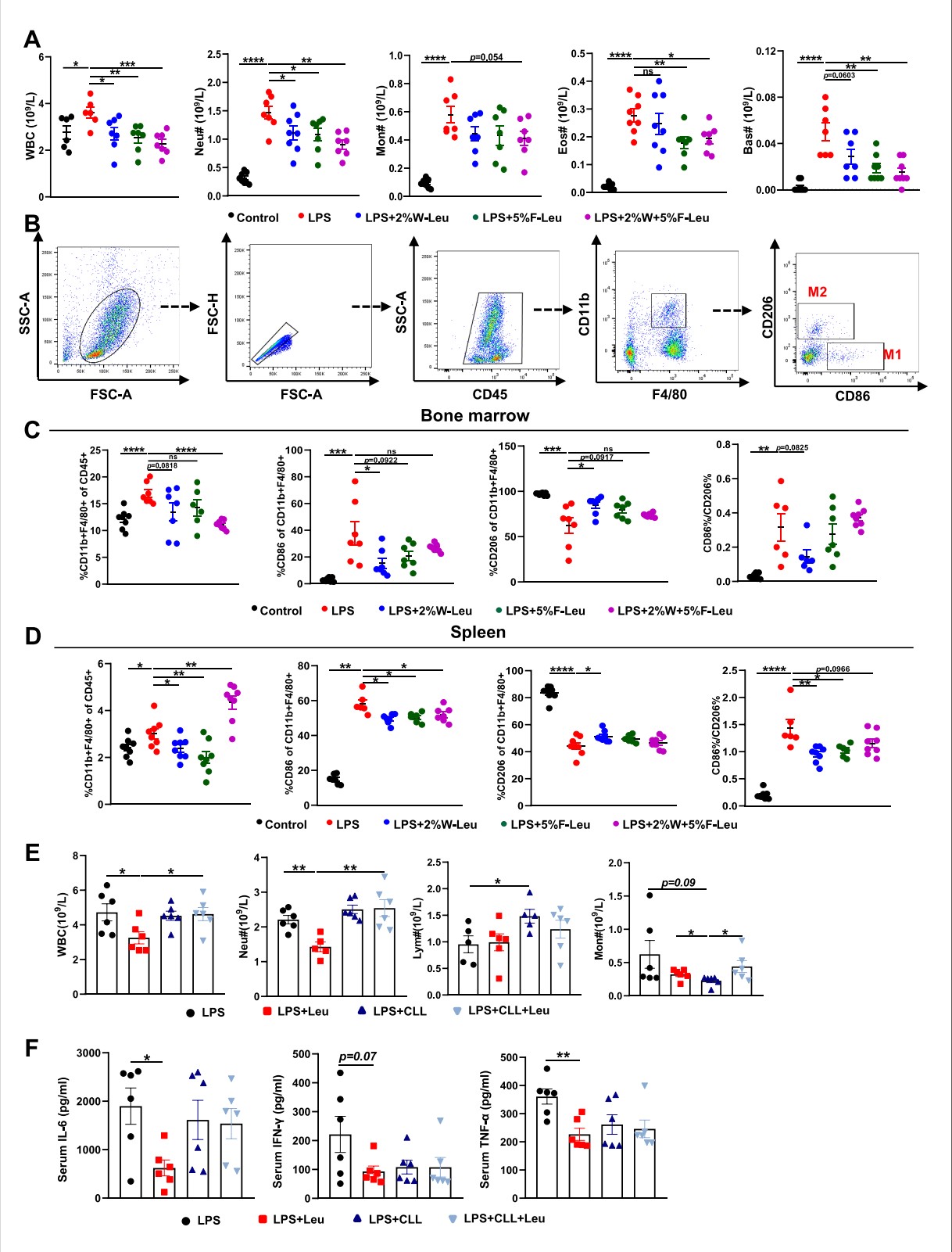

**Figure 2.** Leucine inhibits M1 polarization and promotes M2 polarization in mice. (**A**) White blood cell composition and proportion in mice (*n* = 8). (**B**) Gating strategy for macrophage flow cytometry in the bone marrow. (**C**) Percentages of CD45⁺, CD86⁺, CD206⁺, and CD86⁺/CD206⁺, detected by flow cytometry in the bone marrow (*n* = 8). (**D**) Percentages of CD45⁺, CD86⁺, CD206⁺, and CD86⁺/CD206⁺, detected by flow cytometry in the spleen (*n* = 8). (**E**) White blood cell composition and proportion in mice (*n* = 5–6). (**F**) Measurement of IL-6, IFN-γ, and TNF-α secretion in mouse serum by enzyme-

*Figure 2 continued on next page*

*Figure 2 continued*

linked immunosorbent assay (ELISA) after treatment with lipopolysaccharide (LPS) for 6 hr (*n* = 6). Student's *t*-test was used to determine statistical significance, defined as *p < 0.05, **p < 0.01, ***p < 0.001, and ****p < 0.0001.

The online version of this article includes the following source data for figure 2:

**Source data 1.** Blood biochemical indicators (***Figure 2A***).

**Source data 2.** Flow cytometry of bone marrow macrophages (***Figure 2C***).

**Source data 3.** Flow cytometry of spleen macrophages (***Figure 2D***).

**Source data 4.** Blood biochemical indices in clodronate-containing liposome-treated mice (***Figure 2E***).

**Source data 5.** Serum and peritoneal fluid inflammatory factor levels in clodronate-containing liposome-treated mice (***Figure 2F***).

(***Figure 3B, C***). However, the mRNA expression of *Il6* and *Nlrp3*, which is regulated by NFκB, was uninfluenced by leucine treatment (***Figure 3C***). To rule out that leucine treatment was toxic to cells and thus indirectly regulated the inflammatory response, we examined the effects of 2 and 10 mM leucine on cell viability. The results revealed that cell viability was increased after 6 and 24 hr of 2 and 10 mM leucine treatment, further suggesting that leucine was a direct inhibitor of inflammatory cytokine production (***Figure 3—figure supplement 1A***).

Next, we investigated the effects of leucine on M2 macrophage polarization. IL-4 promoted M2 macrophage polarization, as indicated by increased activity of arginase-1, and induction of mRNA expression of *Arg1*, *Ym1*, *Fizz1*, and *Mgl2* in BMDMs. *Arg1*, a hallmark feature of M2 macrophages, competes with inducible nitric oxide synthase for L-arginine, and decreases nitric oxide synthesis, thereby preventing local inflammation and tissue repair (***Arlauckas et al., 2018***). In IL-4-stimulated cells, leucine increased the activity of Arg1 in the culture supernatant and promoted the protein expression of Arg1, particularly when administered at 10 mM concentration (***Figure 3D–F***). Additionally, leucine promoted the mRNA expression of *Arg1*, *Ym1*, *Fizz1*, and *Mgl2*, thus further validating that leucine promotes M2 polarization (***Figure 3G***). Together, these findings suggested that leucine promotes M2 polarization in BMDMs.

## mTORC1 mediates leucine-induced M2 polarization

M2 polarization involves activation of signal transducer and activator of transcription 6 (STAT6), which directly mediates the transcriptional activation of M2 macrophage-specific genes such as *Arg1* (***Goenka and Kaplan, 2011***). In general, IL-4 activates the STAT6 signaling pathway and consequently promotes Arg1 expression, thus contributing to M2 polarization (***Yang et al., 2021***). However, our results indicated that leucine did not further activate STAT6; therefore, leucine did not promote M2 polarization through the STAT6 pathway (***Figure 4A***). A key effector of leucine is believed to activate the mTORC1 protein kinase (***Cangelosi et al., 2022***). In our study, inhibiting mTORC1 was found to suppress Arg1 expression and inhibit leucine-mediated M2 polarization (***Figure 4A*** and ***Figure 4—figure supplement 1A–C***). Notably, Torin1 inhibited M2 polarization more significantly compared to rapamycin. The above findings underscored the pivotal role of leucine in driving M2 polarization through the mTORC1 pathway.

To further confirm the role of leucine in regulating M2 polarization via mTORC1, we next directly detected the activity of arginase-1 in culture supernatants and cells. Inhibition of mTORC1 was followed by arginase-1 activity inhibition (***Figure 4B,C***). Moreover, the expression of the M2 marker genes *Arg1*, *Fizz1*, *Mgl1*, and *Mgl2* was also completely inhibited (***Figure 4D***). The above results again confirmed that the effect of leucine on M2 polarization occurs through mTORC1 signaling.

In addition, although inhibition of mTORC1 slightly inhibited p-STAT6 activation, it was insufficient to explain the complete inhibition of *Arg1* caused by inhibition of mTORC1, thus indicating that mTORC1 regulates M2 polarization by mediating other pathways. Together, our findings suggested that leucine mediates M2 polarization via the mTORC1 pathway.

## LXRα is essential for leucine-induced macrophage polarization

LXRα is a transcription factor for *Arg1* in macrophages, and its activation enhances the expression of *Arg1* (***Pourcet et al., 2011***). In our study, leucine promoted the protein expression of LXRα upon IL-4 activation of M2 macrophage (***Figure 5A***). The transcriptional regulation of *Arg1* requires the entry of LXRα into the nucleus. Leucine increased the abundance of LXRα in the nucleus (***Figure 5B***) and also

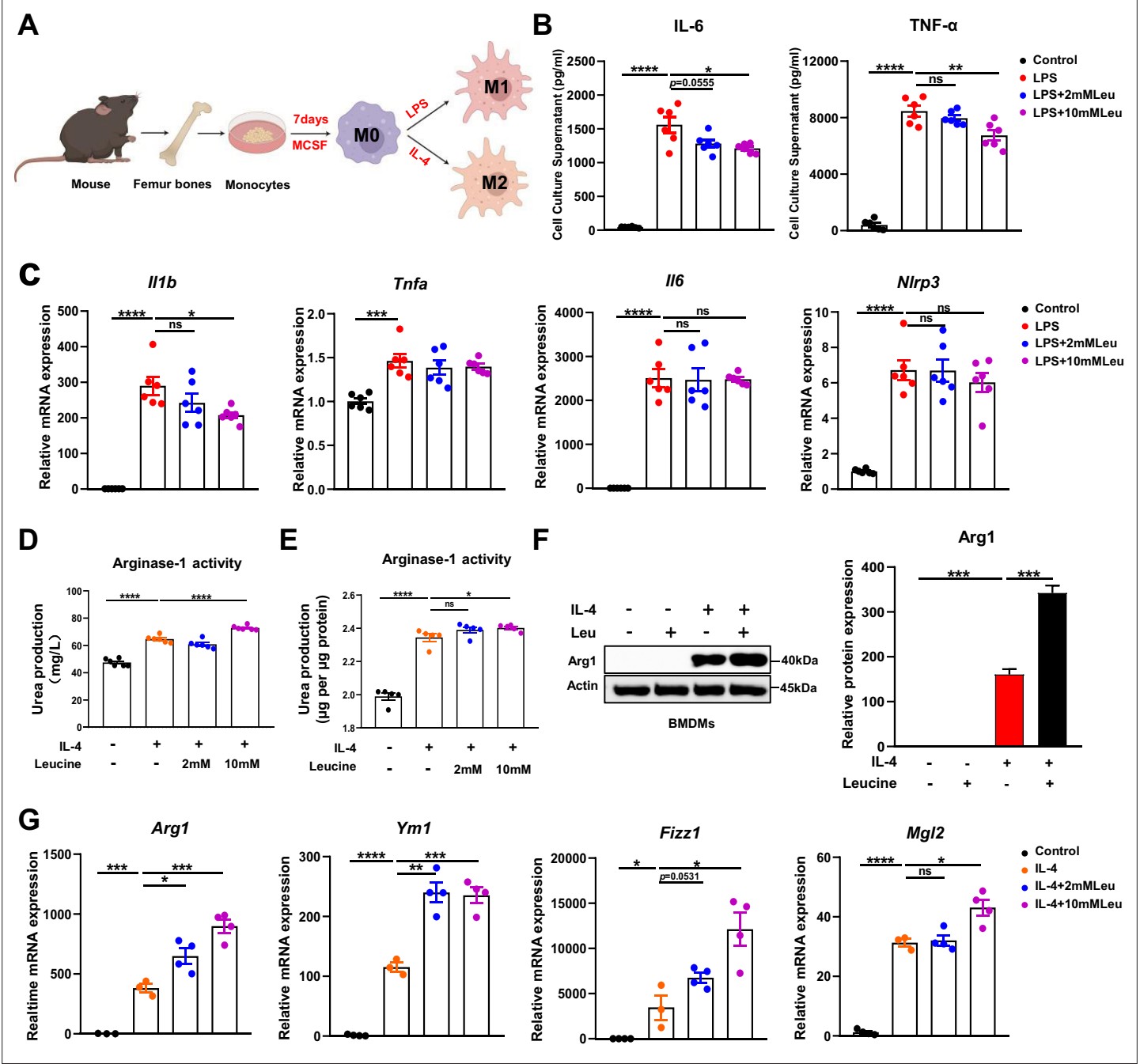

**Figure 3.** Leucine promotes M2 polarization in BMDMs. (**A**) Schematic diagram of macrophage polarization. (**B**) Measurement of IL-6 and TNF-α secretion in cell culture supernatants by enzyme-linked immunosorbent assay (ELISA) (*n* = 6). (**C**) mRNA expression of *Il1b*, *Tnfa*, *Il6*, and *Nlrp3*, measured by real-time PCR in BMDMs (*n* = 6). (**D, E**) Detection of arginase-1 activity in the medium and BMDMs (*n* = 5–6). (**F**) BMDMs isolated from mice were stimulated with leucine, IL-4, or both, and the protein expression of Arg1 was determined. (**G**) mRNA expression of *Arg1*, *Ym1*, *Fizz1*, and *Mgl2*, measured by real-time PCR in BMDMs (*n* = 3–4). Student's *t*-test was used to determine statistical significance, defined as *p < 0.05, **p < 0.01, ***p < 0.001, and ****p < 0.0001.

The online version of this article includes the following source data and figure supplement(s) for figure 3:

**Source data 1.** Levels of inflammatory factors in cell culture supernatants (*Figure 3B*).

**Source data 2.** mRNA expression of inflammatory genes in BMDMs (*Figure 3C*).

**Source data 3.** Arginase-1 activity in culture media and macrophages (*Figure 3D,E*).

**Source data 4.** Original file for the western blot analysis in *Figure 3F*.

*Figure 3 continued on next page*

*Figure 3 continued*

**Source data 5.** mRNA expression of anti-inflammatory genes in BMDMs (*Figure 3G*).

**Figure supplement 1.** Cell viability.

**Figure supplement 1—source data 1.** The raw data for cell viability with different levels of leucine.

increased Arg1 expression (*Figure 5E*) in IL-4-stimulated cells. Moreover, inhibition of LXRα activity by GSK2033 muted the effect of leucine on Arg1 activity and mRNA expression of M2 macrohphage markers *Fizz1*, *Mgl1*, and *Mgl2* demonstrating that LXRα mediates the effect of leucine on M2 polarization (*Figure 5C, D*). Next, we inhibited LXRα in LPS-stimulated mice by i.v. injection of GSK2033, and found that the reduction of cytokine in serum and peritoneal fluid (*Figure 5—figure supplement 1A*). These results showed that LXRα is essential for leucine-induced macrophage polarization.

## Discussion

CSS is an uncontrolled and immune dysregulated immune response involving the sustained activation and expansion of macrophages, which secrete large amounts of cytokines, these cytokines in turn lead to overwhelming systemic inflammation and multi-organ failure with high mortality (*Chen et al., 2020*; *Copaescu et al., 2020*). An imbalance between pro- and anti-inflammatory systems is the main cause of CSS, and macrophage polarization is important for maintaining immune homeostasis. M1 macrophages arise in inflammatory settings dominated by the interferon signaling associated with immunity to bacteria and intracellular pathogens, whereas M2 macrophages relieve inflammation and play important roles in fighting against and recovering from infection (*Murray, 2017*). Thus, therapeutic strategies that target macrophage polarization may be an approach to alleviate CSS.

In our study, leucine decreased LPS-induced inflammation and mortality. The uptake and metabolism of leucine regulated immune cell activation through the mTORC1 signaling pathway. Targeting leucine to manipulate immune responses have been suggested to be useful in the treatment of infections and autoimmunity (*Ananieva et al., 2016*). A previous study has found that leucine inhibits the expression of inflammatory factors in LPS-stimulated RAW 264.7 cells, but the specific mechanism has not been clarified (*Lee et al., 2017*). Our research further verified that leucine alleviates inflammation by inhibiting LPS-stimulated M1 polarization and promoting M2 polarization in both animals and BMDMs.

Macrophage polarization is critical for immune homeostasis. In our study, leucine not only decreased the LPS-mediated production of pro-inflammatory factors such as IL-6, IFN-γ, and TNF-α, but also promoted the expression of the M2 macrophage markers *Mgl1* and *Mgl2*. Alanine aminotransferase (ALT) and aspartate aminotransferase (AST) are sensitive indicators of liver damage. LPS led to significant increase in AST and ALT in both serum and liver, whereas leucine decreased AST and ALT levels, indicating that leucine mitigated tissue damage (*Figure 5—figure supplement 2A*). Importantly, leucine significantly inhibited the high mortality caused by LPS. Therefore, we hypothesized that leucine might regulate macrophage polarization in LPS-induced CSS. Indeed, leucine has been found to directly promote M2 macrophage polarization in vitro, because the addition of leucine markedly increases the expression of M2-associated genes in IL-4-stimulated BMDMs, as well as arginase-1 activity, which is crucial for M2 macrophage polarization (*Zhang et al., 2020a*). Moreover, leucine alone did not increase the expression of M2-associated genes in BMDMs, thus further indicating that leucine synergistically enhances IL-4-induced gene expression and subsequent M2 polarization. CSS is caused by a severe pro-/anti-inflammatory imbalance of macrophages, in which M2, which plays important roles in inflammation relief and tissue repair, is diminished (*Cosín-Roger et al., 2016*). Herein, we demonstrated that leucine alleviates inflammation in LPS-induced CSS, possibly through its ability to promote M2 polarization.

mTORC1 is known to integrate information about cellular nutritional status, including amino acid levels, thus altering cell signaling through a wide range of downstream targets (*Wolfson and Sabatini, 2017*). Recent studies have revealed the crucial importance of the mTORC1 pathway in controlling macrophage polarization, and the strong connections among the regulation of macrophage activity, nutrition sensing, and metabolic status (*Papathanassiu et al., 2017Byles et al., 2013*; *Kobayashi et al., 2021*). Nutrient status, particularly that of amino acids, regulates macrophage polarization via mTORC1 remains unclear. Among amino acids, leucine is a potent activator of mTORC1 in

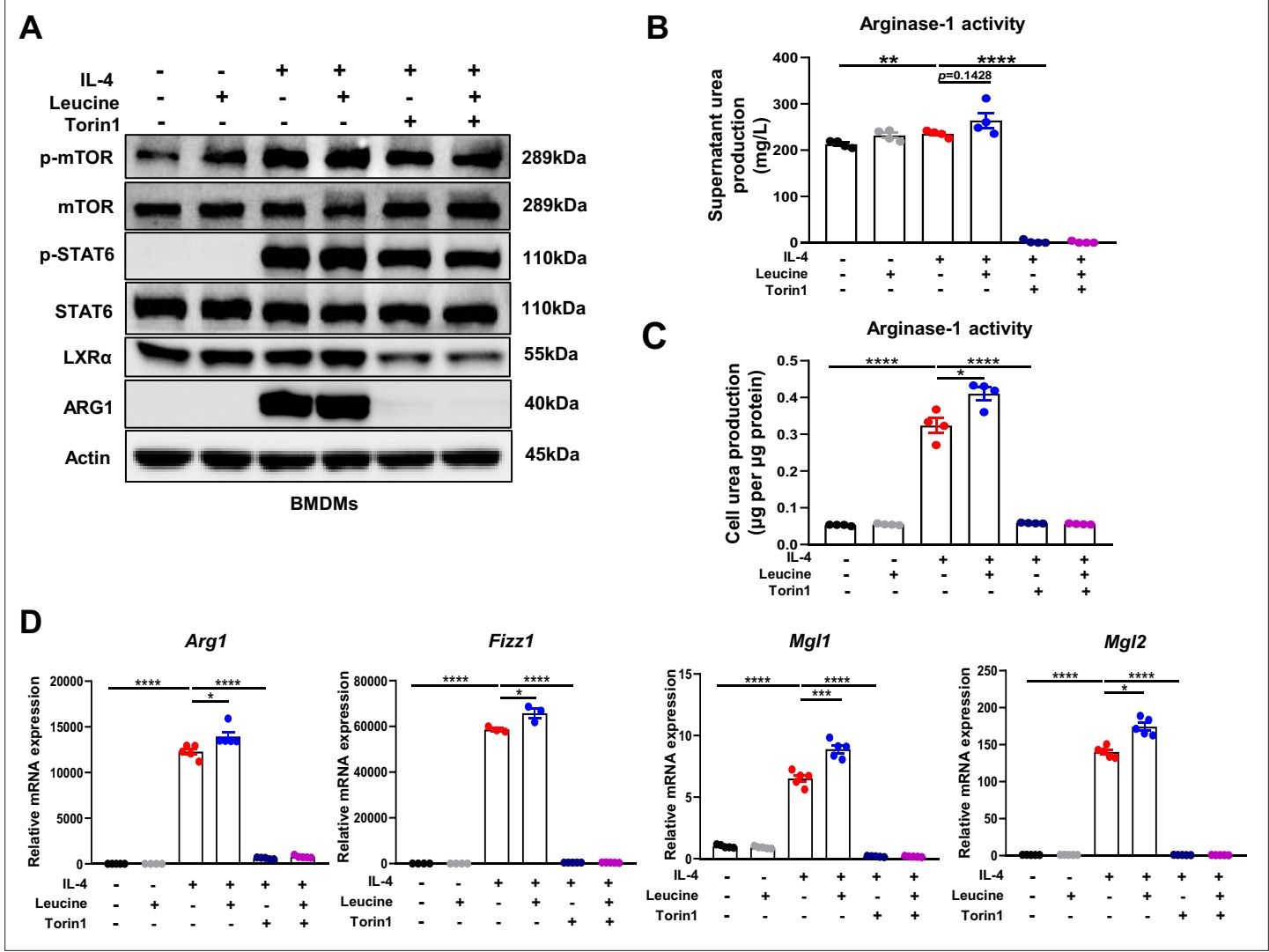

**Figure 4.** Mechanistic target of rapamycin complex 1 (mTORC1) signaling is necessary for M2 polarization. (**A**) Protein levels of ARG1, p-STAT6, STAT6, p-mTOR, and mTOR, determined by western blotting. (**B, C**) Detection of arginase-1 activity in the medium and BMDMs (*n* = 4). (**D**) mRNA expression of *Arg1*, *Fizz1*, *Mgl1*, and *Mgl2*, measured by real-time PCR in BMDMs (*n* = 3–5). Student's *t*-test was used to determine statistical significance, defined as *p < 0.05, **p < 0.01, ***p < 0.001, and ****p < 0.0001.

The online version of this article includes the following source data and figure supplement(s) for figure 4:

**Source data 1.** Original file for the western blot analysis in *Figure 4A*.

**Source data 2.** Urea production in medium (*Figure 4B*).

**Source data 3.** Urea production in BMDMs treated with torin1 (*Figure 4C*).

**Source data 4.** mRNA expression of anti-inflammatory genes in BMDMs treated with torin1 (*Figure 4D*).

**Figure supplement 1.** Inhibition of M2 polarization by rapamycin and wortmannin.

**Figure supplement 1—source data 1.** Inhibition of M2 polarization by rapamycin.

**Figure supplement 1—source data 2.** Inhibition of M2 polarization by wortmannin.

**Figure supplement 1—source data 3.** Torin1 affects the phosphorylation of AKT.

macrophages (*Zhang et al., 2020b*). Leucine activates mTORC1 primarily by blocking the inhibitory effect of the protein setrin2 on the GATOR2, a complex that activates mTORC1 (*Wolfson et al., 2016*). Recently, leucine has been found induce mTORC1 activation in macrophages, thus further regulating macrophage polarization (*Li et al., 2019*).

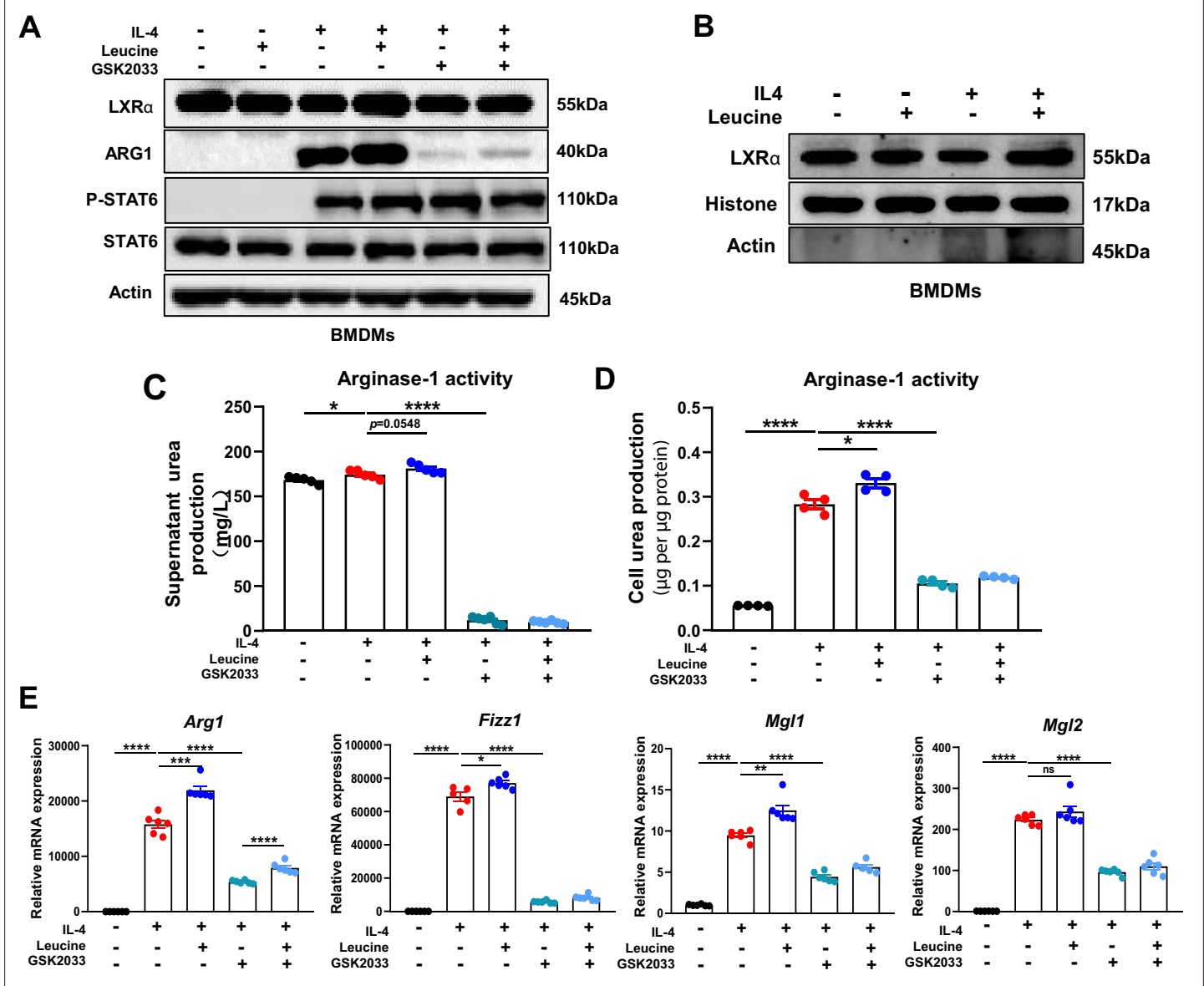

**Figure 5.** Leucine promotes M2 polarization via mechanistic target of rapamycin complex 1 (mTORC1)/liver X receptor α (LXRα) signaling. (**A**) Protein levels of LXRα, ARG1, p-STAT6, and STAT6, determined by western blotting. (**B**) The nuclear proteins of BMDMs were extracted, and the protein levels of histones and LXRα were determined by western blotting. (**C, D**) Detection of arginase-1 activity in the medium and BMDMs (*n* = 4). (**E**) mRNA expression of *Arg1*, *Fizz1*, *Mgl1*, and *Mgl2*, measured by real-time PCR in BMDMs (*n* = 5–6). Student's *t*-test was used to determine statistical significance, defined as *p < 0.05, **p < 0.01, ***p < 0.001, and ****p < 0.0001.

The online version of this article includes the following source data and figure supplement(s) for figure 5:

**Source data 1.** Original file for the western blot analysis in *Figure 5A*.

**Source data 2.** Original file for the western blot analysis in *Figure 5B*.

**Source data 3.** Urea production in medium treated with GSK2033 (*Figure 5B*).

**Source data 4.** Urea production in BMDMs treated with GSK2033 (*Figure 5D*).

**Source data 5.** mRNA expression of anti inflammatory genes in BMDMs treated with GSK2033 (*Figure 5E*).

**Figure supplement 1.** Levels of inflammatory factors in serum and peritoneal fluid of GSK2033-treated mice.

**Figure supplement 1—source data 1.** Levels of inflammatory factors in serum and peritoneal fluid.

**Figure supplement 2.** Aspartate aminotransferase (AST) and alanine aminotransferase (ALT) levels.

**Figure supplement 2—source data 1.** AST and ALT levels.

However, the specific mechanism through which leucine regulates macrophage polarization has not been reported. Akt–mTORC1 signaling integrates metabolic inputs to control macrophage activation. Wortmannin inhibition of AKT was followed by inhibition of M2 polarization, suggesting that AKT signaling is involved in M2 polarization (*Figure 3—figure supplement 1B*). Studies reported that mTORC1 activation inhibits pAkt (T308), inhibition of mTORC1 in turn activate Akt (*Byles et al., 2013*), promoting M2 polarization as a feed back to compensate the inhibition of mTORC1-induced suppression of M2 polarization. mTORC2, directly phosphrlate Akt at S473, and inhibition of mTORC2 inhibits p-Akt (S473) (*Leontieva et al., 2014*), further inhibiting M2 porlarization. Torin1 is the inhibitor for both, while rapamycin is specially for mTORC1 (*Zhao et al., 2015*). In this study, torin significantly decreased p-Akt (S473), and thus additionally inhibits mTORC2 showing a better inhibition of M2 than rapamycin (*Figure 3—figure supplement 1C*).

In M2 macrophages, Arg1 plays a pivotal role in a hallmark characteristic by catalyzing the hydrolysis of L-arginine into L-proline and polyamines, which subsequently downregulate the transcription of pro-inflammatory cytokine TNF-α in macrophages, thereby attenuating local inflammation and promoting tissue repair. Our research findings demonstrate that inhibiting mTORC1 significantly reduces Arg1 expression. Macrophage polarization is regulated by various transcription factors, among which STAT6 is indispensable for M2 polarization. Stimulation of macrophages with IL-4 leads to IL-4R signaling and phosphorylation of the transcription factor STAT6 at tyrosine residues, facilitating its nuclear translocation and induction of target genes. Studies have shown that STAT6/Arg1 is an important signaling mechanism in macrophage phenotypic regulation (*Cai et al., 2019*).Therefore, we hypothesized that leucine might regulate M2 polarization through the mTORC1/STAT6/Arg1 pathway. Indeed, after IL-4 stimulation, STAT6 was tyrosine phosphorylated, but inhibition of mTORC1 did not alter the expression of STAT6. Moreover, leucine increased STAT6 phosphorylation in IL-4-stimulated BMDMs, but leucine alone did not activate STAT6. These findings indicated that leucine promotes macrophage M2 polarization independently of the mTORC1/STAT6/Arg1 pathway.

Beyond STAT6, a prior study has identified that LXRα regulates Arg1 expression in macrophages by promoting binding of the hematopoietic transcription factors IRF8 and PU.1 to the transcription start site in the Arg1 gene. The importance of LXRα in inhibiting the expression of inflammatory mediators in macrophages and macrophage differentiation has been reported (*A-Gonzalez et al., 2013*; *Joseph et al., 2003*). LXRα knockout decreases Arg1 expression, thus enhancing inflammation signatures of macrophages and ultimately inhibiting recovery after injury (*Mao et al., 2021*). However, whether leucine regulates M2 polarization through mTORC1/LXRα must be verified. After inhibition of mTORC1, the protein expression of LXRα decreased, and Arg1 expression was also significantly inhibited. Moreover, after LXRα inhibition, the expression of Arg1 decreased significantly, thereby confirming that active LXRα is necessary for M2 polarization. These findings indicated that leucine-mediated M2 polarization may occur via the mTORC1/LXRα/Arg1 pathway. Indeed, we extracted macrophage nuclear proteins and found that leucine effectively promoted LXRα entry into the nucleus. Thus, our results showed that mTORC1 integrates intracellular leucine signaling and external IL-4 signaling, thus activating its downstream transduction factor LXRα, and promoting LXRα entry into the nucleus and the induction of the target gene *Arg1*, and ultimately leading to M2 polarization.

Our results suggested that leucine regulates M2 via the mTORC1/LXRα/Arg1 pathway, thus alleviating LPS-mediated CSS. Leucine also has regulatory effects on M1. In present study, we report the first evidence that leucine decreases mortality in mice after a lethal dose of LPS, and attenuates secretion of the pro-inflammatory factors IL-6, IFN-γ, and TNF-α in the serum. The anti-inflammatory effects of leucine were probably mediated by modulation of macrophage polarization, because leucine suppressed CD86 expression (M1 macrophage marker) but increased CD206 (M1 macrophage marker) expression in both the bone marrow and spleen. This possibility was corroborated by in vitro data in BMDMs. Leucine inhibited LPS-driven M1 polarization, and decreased the secretion of IL-6, IL-1β, and TNF-α in BMDMs. Together, our in vivo and in vitro results suggest that leucine may also inhibit inflammation driven by M1 macrophages. Therefore, subsequent studies of the specific mode of action of leucine on the polarization of M1 macrophages will be essential.

## Conclusions

In summary, the present study revealed that leucine ameliorates CSS in mice exposed to LPS by inhibiting macrophage M1 polarization and promoting M2 polarization. On the basis of our results, a

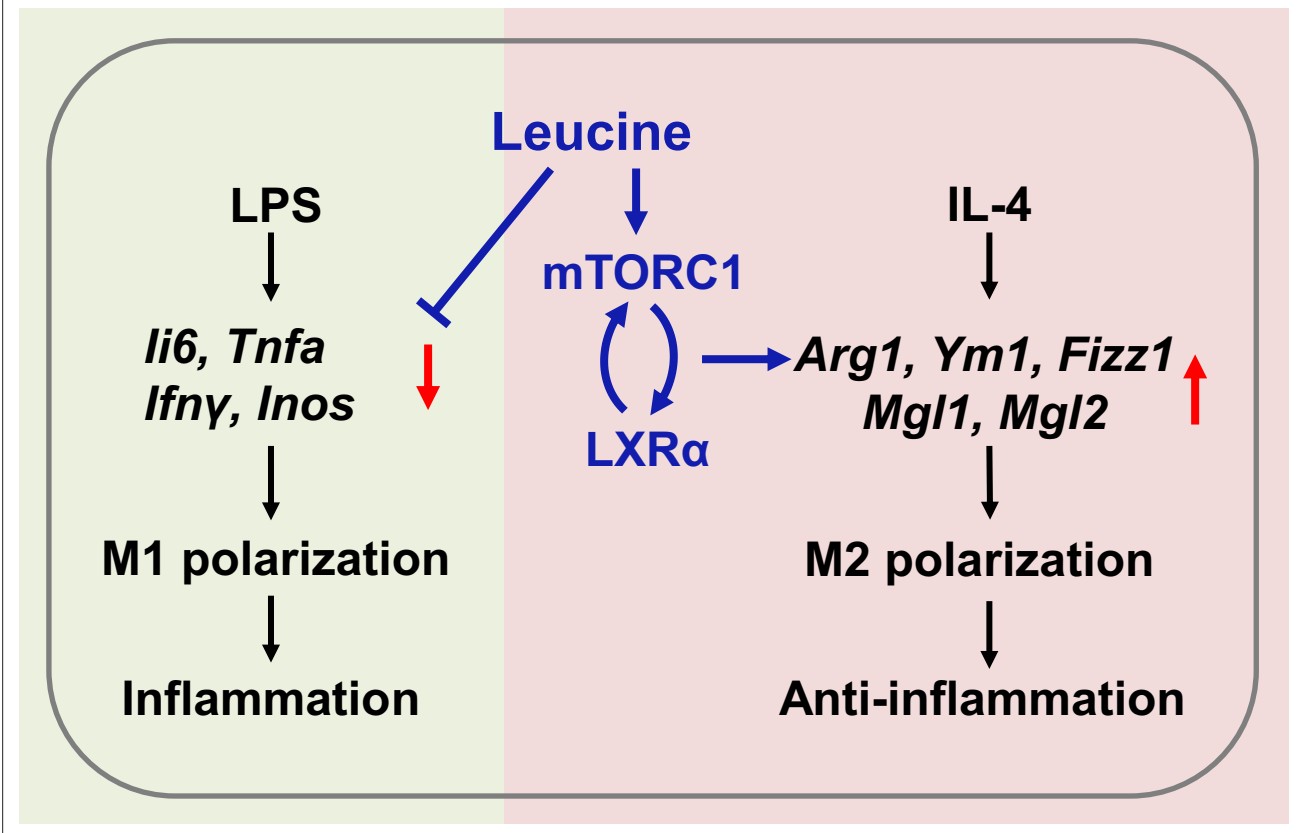

**Figure 6.** Mechanism of leucine alleviating lipopolysaccharide (LPS)-induced cytokine storm syndrome (CSS) by modulating mechanistic target of rapamycin complex 1 (mTORC1)/liver X receptor α (LXRα) signaling. In macrophages, LPS promotes M1 polarization to promote the secretion of inflammatory factors leading to inflammation, and IL-4 promotes M2 polarization to alleviate inflammation. Leucine further promotes IL-4-induced M2 polarization by activating mTORC1/LXRα to alleviate inflammation and repair damaged tissues, while leucine also inhibits LPS-mediated M1 polarization and reduces the expression and secretion of inflammatory factors in the organism.

role of leucine in macrophage inflammatory responses via the mTORC1/LXRα/Arg1 axis is proposed (*Figure 6*), in which leucine promotes M2 macrophage polarization through the mTORC1/LXRα/Arg1 signaling pathway, thereby contributing to the resolution of inflammation and the repair of damaged tissues.

## Materials and methods
### Animals
The experimental procedures and animal care were performed in accordance with the regulations of the Animal Care Committee of Sichuan Agricultural University (No. 20180701). Forty male C57BL/6J mice (8 weeks) were purchased from Dashuo Laboratory Animal Co, Ltd (Chengdu, China). Mice with similar body weights (n = 8) were randomly divided into five treatment groups: (1) control group; (2) LPS group; (3) LPS + 2% leucine drinking water group (LPS + 2% W Leu); (4) LPS + 5% leucine food group (LPS + 5% F Leu); and (5) LPS + 2% leucine drinking water + 5% leucine food group (LPS + 2% W + 5% F Leu). All mice were housed in cages with constant humidity (40–70%) and temperature (20–25°C) under a 12-hr light/dark cycle, and were given free access to drinking water and food for 21 days. On day 21, the mice were challenged with LPS (intraperitoneal injection); 6 hr after the challenge, the mice were anesthetized by 20 s exposure to carbon dioxide, and blood samples were collected through cardiac puncture. Collected samples were snap frozen in liquid nitrogen and stored at −80°C until analysis. Blood samples were centrifuged at 3000 × *g* for 15 min at 4°C, and then the serum was separated and stored at −20°C until further analysis.

## Macrophage isolation and stimulation

BMDMs were prepared as previously described (*Byles et al., 2013*). Briefly, after 6- to 8-week-old C57BL/6J mice were euthanized with $CO_2$, the femurs and tibias were removed and centrifuged to obtain cells. For macrophage differentiation, bone marrow-derived cells were plated in Petri dishes and cultured for 7 days in αMEM (containing 10% fetal bovine serum and 1% penicillin/streptomycin) supplemented with 10 ng/ml macrophage colony-stimulating factor (M-CSF). Adherent cells were collected and seeded into new dishes for subsequent experiments. For M1-like activation, $25 \times 10^4$ BMDMs were placed in 12-well plates and treated with 100 ng/ml LPS (Sigma–Aldrich) for 6 hr. For M2 polarization, cells were treated with 20 ng/ml IL-4 (Peprotech) for 24 hr. Leucine treatment was usually performed 1 hr before LPS/IL-4 stimulation.

## Real-time quantitative PCR

Total RNA was extracted with TRIzol reagent (Invitrogen) according to the manufacturer's instructions. The reaction solution was prepared according to the instructions of the reverse transcription kit (Takara) to reverse-transcribe RNA to cDNA. QPCR was then performed with a reaction mixture consisting of 5 μl SYBR Green (Takara), 0.2 μl Rox, 3 μl $dH_2O$, 0.4 μl primers (F +R) for each gene used in the study, and 1 μl cDNA. Relative gene expression was calculated with the ΔΔCT method, and results were normalized to values for the housekeeping gene *Ppia*. Primer sequences are listed in *Supplementary file 1*.

## Western blot analysis

Cells were washed with ice-cold phosphate-buffered saline, and proteins were extracted with RIPA lysis buffer (containing phenylmethanesulfonylfluoride (PMSF) and phosphatase inhibitors). After a 30 min incubation at 4°C, samples were sonicated and centrifuged at $12,000 \times g$ for 15 min at 4°C, and clear supernatant was collected. Concentrations were determined, and samples were assayed with a BCA protein assay kit (Thermo Scientific, MA, USA). Equal amounts of protein were then separated through 10% sodium dodecyl sulfate–polyacrylamide gel electrophoresis and transferred to polyvinylidene fluoride membranes (Merck Millipore Ltd, Tullagreen, Ireland). Membranes were blocked in 5% nonfat dry milk in 1× tris buffered saline with Tween 20 (TBST) for 1 hr at room temperature, then incubated with specific primary antibodies overnight at 4°C. Membranes were washed three times with TBST and incubated with horseradish peroxidase (HRP)-conjugated secondary antibodies for 1 hr at room temperature. Finally, protein bands were visualized with an ECL chemiluminescence kit (Beyotime Biotechnology, Shanghai, China). Protein band density was quantified in Image Lab software (Bio-Rad). The ratio of the densitometric values of the target protein to the reference protein was calculated and expressed relative to the control value. Antibody information is listed in *Supplementary file 2*.

## Flow cytometry analysis

Fluorescently labeled antibodies (purified anti-mouse CD16/32, FITC anti-mouse CD45, APC anti-mouse F4/80, PerCP/Cy5.5 anti-mouse CD11b, APC/Cy7 anti-mouse CD86, and PE/Cy7 anti-mouse CD206) were used according to the manufacturer's instructions. Cells were collected on a BD FACSVerse instrument (BD Biosciences) and analyzed in FlowJo10 software. Antibody information is listed in *Supplementary file 2*.

## Arginase assays

Arginase was measured as described above (*Corraliza et al., 1994*). In brief, cells were lysed with 0.1% Triton X-100 and incubated at 37°C for 30 min to release enzymes by cell rupture. Subsequently, $MnCl_2$ and Tris–HCl (pH = 7.5) were added and heated at 56°C for 10 min to activate arginase-1. Subsequently, 500 mM L-arginine (pH = 9.7) was added and incubated at 37°C for 30 min to hydrolyze L-arginine. Hydrolysis was stopped with acid stop solution ($H_2SO_4:H_3PO_4:H_2O$ = 1:3:7 vol/vol). Finally, 9% α-isonitrosopropiophenone (dissolved in 100% ethanol) was added and heated at 100°C for 15 min. Urea was measured at 540 nm, and all samples were read in triplicate.

## Enzyme-linked immunosorbent assays

The cytokines TNF-α, IL-6, and IFN-γ (Beijing Sizhengbai Biotechnology, China) were determined with commercial enzyme-linked immunosorbent assays. Briefly, serum or cell culture supernatants were collected and analyzed according to the manufacturer's recommendations.

## AST and ALT assays

For liver function tests, glutamic oxaloacetic transaminase (also known as aspartate transaminase, AST) and ALT levels in mouse serum and liver homogenate supernatants were detected with kits (Nanjing Jiancheng Bioengineering Institute, Nanjing, China) according to the manufacturer's instructions.

## Routine blood examination

Analysis of white blood cell composition and proportion (WBC, Neu#, Mon#, Lym#, Bas#, Eos#) was performed with an automatic biochemical analyzer (Hitachi 3100).

## Statistical analysis

The results are presented as mean ± standard error of the mean. Groups were compared with unpaired two-tailed Student's $t$-test and/or one-way analysis of variance. The p values are indicated in the figures as follows: $*p < 0.05$, $**p < 0.01$, $***p < 0.001$, $****p < 0.0001$, and ns, not significant ($p > 0.05$). All results were plotted in GraphPad Prism 8 software.

## Acknowledgements

This work was supported by the Sichuan Science and Technology Program under grants (2021YJ0195, 2021ZDZX0009, and 2020YFN0147).

## Additional information

### Funding

| Funder | Grant reference number | Author |
| --- | --- | --- |
| National Key Research and Development Program of China | 2023YFD1300803 | Hui Yan |
| Natural Science Foundation of China | U22A20513 | Daiwen Chen |
| Science and Technology Program of Sichuan | 2021ZDZX0009 | Hui Yan |
| Science and Technology Program of Sichuan | 2021YJ0195 | Hui Yan |

The funders had no role in study design, data collection, and interpretation, or the decision to submit the work for publication.

### Author contributions

Hui Yan, Conceptualization, Resources; Yao Liu, Conceptualization, Data curation, Validation, Investigation, Visualization, Methodology, Writing – original draft, Project administration, Writing – review and editing; Xipeng Li, Writing – original draft; Bing Yu, Xiangbing Mao, Zhiqing Huang, Junqiu Luo, Methodology; Jun He, Yuheng Luo, Visualization; Jie Yu, Project administration; Aimin Wu, Writing – review and editing; Daiwen Chen, Resources, Methodology, Writing – original draft, Writing – review and editing

### Author ORCIDs

Hui Yan https://orcid.org/0000-0001-7476-3914
Yao Liu https://orcid.org/0000-0001-5141-3161
Zhiqing Huang https://orcid.org/0000-0001-5092-9297
Daiwen Chen https://orcid.org/0000-0002-8351-7421

### Ethics

This study was performed in strict accordance with the recommendations in the Guide for the Animal Care Committee of Sichuan Agricultural University. The protocol was approved by the Committee on the Ethics of Sichuan Agricultural University (Permit Number: 20180701). All surgery was performed under sodium pentobarbital anesthesia, and every effort was made to minimize suffering.

Joint Public Review https://doi.org/10.7554/eLife.89750.3.sa1
Author Response https://doi.org/10.7554/eLife.89750.3.sa2

## Additional files

### Supplementary files

• MDAR checklist

• Supplementary file 1. qPCR primer sequences.

• Supplementary file 2. Antibody information.

### Data availability

All data generated or analyzed during this study are included in the manuscript and supporting files.

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
