## [Editor Report · eLife assessment]

The study has added value to what we have already known in the potential pharmacological immunomodulatory therapies in LPS-induced sepsis, and especially the use of oral leucine might be of great interest to the readers engaged in this field. We believe this study is **important** and provides **solid** evidence on the potential use of leucine in sepsis.

---

## [Referee Report · Joint Public Review]

Summary:

The major purpose of this manuscript is to examine whether leucine treatment would be a potential strategy to treat cytokine storm syndrome (CSS). CSS is a common symptom in multiple infectious diseases in clinic, gradually leads to multiple organ failure and high mortality. Strategies to treat CSS including pulse steroid therapy normally leads to severe side effects. Therefore, it is still required to develop safe strategy with high efficacy to treat CSS. In clinic, sepsis is well characterized to exhibit CSS and therefore multiple studies utilized LPS-induced sepsis model to evaluate CSS symptom. In this study, the authors examined whether leucine, an essential amino acid that has been absorbed daily in our body, could ameliorate CSS symptom in the LPS-induced sepsis mouse model. They found a potential protective effect of leucine in terms of the survival rate and inflammatory responses.

Strengths:

The study is overall well designed and the results are well analyzed with only minor issues. The methods they utilized is appropriate.

Weaknesses:

The mechanistical insights are not sufficient and could not fully explain the phenotype they found. Considering the importance of this study is to identify the potential protective role of leucine in CSS, the authors could also consider investigator-initiated clinical trials to further expand the significance of this study.

---

## [Author Response]

The following is the authors’ response to the current reviews.

I greatly appreciate your time and attention on our manuscript. I have carefully considered the reviewers’ comments and made modifications. Below are my responses to each comment and the revisions I have made.

**Reviewer #2 (Recommendations for The Authors):**
1. The authors address well with most of my concerns. I am fine with most of the responses except question 8. Actin is also reported to be located in nuclear (PMID: 31481797). It would be better to utlize other markers, like GAPDH. Moreover, the author did not address the issue of LXRa. I strongly suggest that the authors repeat this experiment to get a more solid result.

Thank you for the comment! Actin is frequently used as a negative control for nucleus protein in many publications, such as DOI:10.1038/s41419-018-0428-x. Beta-actin is rich in cytoplasm protein that it only takes few seconds to reveal the strong band when performing western blot with cytoplasm. However, actin does not reveal when exposing western- blot with nucleus for minutes in many studies, including in this study. Even though as mentioned actin is also located in the nuclear, such a tiny amount in the nucleus may not be revealed in western blot with exposure in seconds. However, if nucleus protein is contaminated with total cell lysate, the action is quite easy to reveal. As a result, the use of actin as the nagtive control of nucleus protein is well-accepted.

**Author response image 1. sa2fig1:** 

1. In addition, the authors mentioned IL-1b but present IL-6 in the figure of Figure. 2F. Please correct.

We appreciate your attention on the detail. “IL-1b” is corrected to “IL-6”.

The following is the authors’ response to the original reviews.

I greatly appreciate the time you and the reviewers have taken to review my paper and provide detailed feedback and suggestions. I have carefully considered the reviewers’ comments and made thorough modifications to the paper. Below are my responses to each comment and the revisions I have made.

**Reviewer #1 (Recommendations for The Authors):**
Although the paper has strengths in understanding better the pathway of activation leading to polarization, the mechanisms contributing to cytokine storm are weak. In the context of cellular in vitro changes, it would be very interesting to map these molecular changes to strengthen the pathways affected in this model. In vivo, stronger evidence is required to bridge the gap between the in vitro model and mechanisms regulating in vivo disease development. Reporting of experiments needs to be considerably strengthened. Individual data points are shown, however, it is unclear whether these represent biological or technical, or how many experiments have been undertaken. The addition of this information is essential for uznderstanding the robustness and repeatability of findings. Currently, these cannot be assessed from the information provided. Furthermore, it is unclear whether the error bars represent s.e.m or s.d. which greatly impacts data interpretation.

Answer: thank you for the valuable comments! We have added some in vivo experiments to strengthen the bridge between the in vitro and in vivo model. (1) The depletion of macrophage by clodronate-liposomes (CLL) i.v. injection was performed in endotoxemic mice with leucine. The alleviation of LPS-induced cytokine production by leucine was muted with macrophage depletion (Figure 2E, F), suggesting the anti-inflammatory effect of leucine was exerted via the regulation of macrophage. (2) The LXRα inhibitor, GSK2033, was applied to mice via i.v. injection prior to LPS-challenge. In GSK2033 treated mice, the effects of leucine on the serum levels of inflammatory cytokines were neutralized (Supplementary Figure 4), partially indicating the importance of LXRα in the regulation of cytokine release. We acknowledge the limitation of LXRα inhibition by GSK2033 in this study. In our future study, we plan to use monocyte specific LXRα knockout mice by LysM-cre to elucidate the importance of LXRα in the progression of CSS, and specifically focuse on the molecular mechanism how mTORC1 interacts with LXRα to modulate M2 macrophage polarization. Additionally, we made modifications in the manuscript to clarify that the error bars represented as the standard error of the mean (SEM) (line 416).

**Reviewer #2 (Recommendations for The Authors):**
1. The whole manuscript is based on the 2% leucine from feed and 5% leucine from water. Is there any rationale for using these two types of different concentrations in this study? Often, a dose-dependent treatment is utilized in vivo in pharmacological study. Therefore, the authors should at least test two different concentrations in each type to confirm the conclusion.

Answer: thank you for your comment and suggestion. The 2% leucine in feed and 5% leucine in water in this study were based on the literatures. In those studies, leucine was reported to activate mTORC1 and regulate metabolism at such types of different concentration as shown below, although there is lack of leucine in the regulation of macrophage activation. In this study, we found leucine supplementation in such types significantly increased the average body weight gain of mice, suggesting growth promoting and no toxicity of leucine on mice.

(1) Jiang X, Zhang Y, Hu W, Liang Y, Zheng L, Zheng J, Wang B, Guo X. 2021. Different Effects of Leucine Supplementation and/or Exercise on Systemic Insulin Sensitivity in Mice. Front Endocrinol (Lausanne) 12:651303. doi:10.3389/fendo.2021.651303

(2) Holler M, Grottke A, Mueck K, Manes J, Jücker M, Rodemann HP, Toulany M. 2016. Dual Targeting of Akt and mTORC1 Impairs Repair of DNA Double-Strand Breaks and Increases Radiation Sensitivity of Human Tumor Cells. PLoS One 11: e0154745. doi:10.1371/ journal. pone.0154745

1. The authors focus on macrophage polarization as the major cellular event affected by leucine treatment; however, they also report that the proportion of multiple immune cell types has been suppressed by leucine treatment. As some of these immune cells can also produce inflammatory cytokines, the authors should confirm the anti-inflammatory effects of leucine were mainly mediated by modulating macrophage polarization as they suggested in the manuscript. For example, the authors could utilize Anti-CSF1 or clodronate to deplete macrophage and observed whether leucine-reduced inflammatory cytokines production was largely diminished.

Answer: thank you for your valuable suggestion! We used clodronate-liposome (CLL) i.v. injection to deplete macrophages to further validate the specific contribution of macrophage polarization to the anti-inflammatory effects of leucine. The results revealed that clodronate treatment decreased blood monocyte counts and eliminated the effect of leucine in lowering serum inflammatory factors IL-6, IFN-γ and TNF-α (Figure 2E-F), suggesting the importance of leucine-mediacted macrophage activation on the anti-inflammation.

1. It would be important to examine whether 10 mM leucine would exhibit cytotoxicity to bone marrow derived monocytes/macrophages. This would confirm that leucine treatment directly suppresses inflammatory cytokines production or reduces cell viability to indirectly modulates inflammatory responses.

Answer: thank you for your valuable suggestion! We performed cell viability assays after treating BMDM with 2 mM and 10 mM leucine for 6h or 24h (consistent with the timing of leucine treatment in article). The results showed that at 6h, 2 mM leucine significantly increased cell viability, while 10 mM leucine had no significant effect on cell viability. At 24h, both 2 mM and 10 mM leucine significantly increased cell viability. In conclusion, 2 mM and 10 mM leucine were not cytotoxic to BMDM, and the anti-inflammatory effect of leucine was not derived from the reduction in cell viability (Supplementary Figure 2).

1. The authors found that leucine promotes mTORC1-LXRα for arginase-1 transcription and M2 polarization. The pathway the authors elucidated is not surprising, which has already been reported in other studies. What about the other M2 markers? The authors could examine whether arginiase-1 deficiency would deplete leucine-increased other M2 marker genes expression. Moreover, what about the molecular mechanism for leucine-reduced M1 polarization?

Answer: Thank you for the valuable comments! To clarify that Arginase-1 activity, mRNA expression of Fizz1, Mgl1, Mgl2, and Ym1 were well established markers for M2 macrophage. Specifically, Arginase-1 activity is important to define M2 functionality. These markers were used to define the level of M2 macrophage polarization. Only a few studies indicated the involvement of mTORC1 in the M2 polarization as shown below; however, there is no molecular mechanism about how mTORC1 modulates this process. In this study, we provide the evidence that LXRα mediated the mTORC1 associated M2 polarization, and leucine regulated mTORC1-LXRα to promote M2 polarization, which was in dependent of IL-4-induced STAT6 signaling. In our future study, we are focusing on the molecular mechanism how mTORC1 interacts with LXRα to modulate M2 macrophage polarization.

(1) Byles V, Covarrubias AJ, Ben-Sahra I, Lamming DW, Sabatini DM, Manning BD, Horng T. 2013. The TSC-mTOR pathway regulates macrophage polarization. Nat Commun 4:2834. doi:10.1038/ncomms3834

(2) Kimura T, Nada S, Takegahara N, Okuno T, Nojima S, Kang S, Ito D, Morimoto K, Hosokawa T, Hayama Y, Mitsui Y, Sakurai N, Sarashina-Kida H, Nishide M, Maeda Y, Takamatsu H, Okuzaki D, Yamada M, Okada M, Kumanogoh A. 2016. Polarization of M2 macrophages requires Lamtor1 that integrates cytokine and amino-acid signals. Nat Commun 7:13130. doi:10.1038/ncomms13130

1. In Fig. 1A, what's the P-value among these two groups? Moreover, what about the result with combination treatment as the authors performed in other panels?

Answer: thank you for the valuable comments from the reviewer! In Figure 1A, the P-value between the LPS and LPS+2% Leucine groups is 0.0031, and the P-value between the LPS and LPS+5% Leucine groups is 0.0009. I have marked the significance in Figure 1A accordingly. Due to the limited number of mice, we only treated mice in two different ways respectively. Initially, we performed survival experiment and observed that the addition of leucine prolonged survive of mice at lethal dose. Based on these findings, we further investigated whether a combination of the two methods would yield better results on the regulation of inflammation, but the combination exhibited the similar effect on cytokines production, and it is not necessary to repeat the survival experiment with the combination.

1. It seems not much difference could be observed between 2% leucine from feed and 5% leucine from water in the expression of inflammatory genes and anti-inflammation-related markers. However, it seems that 5% leucine from water would exhibit a better survival rate than 2% leucine from feed. The authors should explain potential reasons and at least examine it in vitro.

Answer: we appreciate the valuable comments from the reviewer! There are two possible reasons: (1) When lethal dose of LPS applied, mice were too weak to eat but still drank a small amount of water; (2) the absorption of leucine from the water were much easier than from the feed, thus leucine from the water exhibited much better efficiency in a short period of survival experiment. On the other hand, the cytokine levels and expressions were measure in non-lethal experiments, in which mice were in much better condition for lecine absorption.

1. In Fig. 4A, the authors examined the expression of p-mTOR. The authors should further examine the expression of p-AKT (S473, T308) and p-S6 to clarify whether mTORC1 or mTORC2 has been modulated. As reported, leucine should act on GATOR2 for mTORC1 activation. However, the authors reported that Torin, a mTORC1/mTORC2 inhibitor, inhibited M2 polarization more significantly compared to rapamycin, a mTORC1 inhibitor. These observations seem to indicate that leucine has other targets except mTORC1, such as mTORC2, which might raise novel mechanisms that have never been reported before.

Answer: thank you for the valuable comments! Akt-mTORC1 signaling integrates metabolic inputs to control macrophage activation. Wortmannin inhibition of AKT was followed by inhibition of M2 polarization, suggesting that AKT signaling is involved in M2 polarization. Studies reported that mTORC1 activation inhibits pAkt (T308), inhibition of mTORC1 in turn activate Akt (1), promoting M2 polarization as a feed back to compensate the inhibition of mTORC1 induced suppression of M2 polarization. mTORC2, directly phosphrlate Akt at S473, and inhibition of mTORC2 inhibits p-Akt (S473) (2), further inhibiting M2 porlarization. Torin1 is the inhibitor for both, while rapamycin is specially for mTORC1 (3). The explanation was included in Line 252-262

(1) Leontieva OV, Demidenko ZN, Blagosklonny MV. 2014. Rapamycin reverses insulin resistance (IR) in high-glucose medium without causing IR in normoglycemic medium. Cell Death Dis 5: e1214. doi:10.1038/cddis.2014.178Byles.

(2) Holler M, Grottke A, Mueck K, Manes J, Jücker M, Rodemann HP, Toulany M. 2016. Dual Targeting of Akt and mTORC1 Impairs Repair of DNA Double-Strand Breaks and Increases Radiation Sensitivity of Human Tumor Cells. PLoS One 11: e0154745. doi:10.1371/journal. pone .0154745

(3) V, Covarrubias AJ, Ben-Sahra I, Lamming DW, Sabatini DM, Manning BD, Horng T. 2013. The TSC-mTOR pathway regulates macrophage polarization. Nat Commun 4:2834. doi:10.1038/ncomms3834.

1. In Fig.5B, frankly speaking, I do not observe much difference in LXRα expression. Also, the actin band is too poor to get any conclusion.

Answer: thank you for the valuable comments from the reviewer! In Fig. 5B, the extracted protein is specifically mentioned as nuclear protein in the text. It is stated that actin is expressed in the cytoplasm, while histone is expressed in the nucleus. The figure shows that actin expression is almost absent, which is mentioned to demonstrate the purity of the extracted nuclear protein.

1. In Fig. 5C and 5D, it is amazing that GSK2033 would reduce urea production even largely greater than the basal condition (lane 1). As GSK2033 normalized IL-4 or IL-4 combination with Leucine raised urea production in cells, how GSK2033 could reduce urea in medium. The authors should explain this discrepancy.

Answer: thank you for the valuable comments from the reviewer! In Fig. 5C, urea production was measured directly in the culture medium using a commercial assay kit, and GSK2033 indeed led to a significant decrease in urea production. In Fig. 5D, on the other hand, we assessed the activity of arginase-1 by lysing the cells, activating arginase-1, providing the substrate arginine, and then measuring urea production. In response to your question, the explanation is that in the assay measuring arginase-1 activity, we supplied a sufficient amount of substrate arginine, which may better reflect the enzyme’s activity and the results were consistent with our expectations. Additionally, when GSK2033 was used in combination with IL-4 or IL-4 plus leucine, it might interact with the IL-4 signaling pathway or leucine metabolism pathway, leading to an increase in urea production. This is just our preliminary explanation for the contradictory results, and we acknowledge that further research is needed to explore the mechanism of action of GSK2033 and its interactions with IL-4 or leucine.

1. Line 98, "INF-gamma" should be IFN-gamma.

Answer: We appreciate your attention to detail. We apologize for the error in line 98, where “INF-gamma” should indeed be corrected to “IFN-gamma (IFN-γ).” We will make the necessary correction in the revised version of the manuscript.